# Levels of Somatic Anxiety, Cognitive Anxiety, and Self-Efficacy in University Athletes from a Spanish Public University and Their Relationship with Basic Psychological Needs

**DOI:** 10.3390/ijerph20032415

**Published:** 2023-01-29

**Authors:** Isabel Mercader-Rubio, Nieves Gutiérrez Ángel, Sofia Silva, Sónia Brito-Costa

**Affiliations:** 1Department of Psychology, Faculty of Education Sciences, Universidad de Almería, 04120 Almería, Spain; 2Polytechnic of Coimbra, Coimbra Education School, Research Group in Social and Human Sciences (NICSH), 3030-329 Coimbra, Portugal; 3Polytechnic of Coimbra, Institute of Applied Research (i2A), 3030-329 Coimbra, Portugal; 4Polytechnic of Coimbra, Human Potential Development Center (CDPH), 3030-329 Coimbra, Portugal

**Keywords:** somatic anxiety, cognitive anxiety, self-efficacy, basic psychological needs, university students

## Abstract

Research on self-efficacy, cognitive anxiety and somatic anxiety provides fundamental information to early identify weak areas in the training of athletes and to implement actions that contribute to the improvement and maintenance of sporting activities. The aim of this study was to analyze the relationship between anxiety (somatic anxiety, cognitive anxiety and self-efficacy) and basic psychological needs (competence, autonomy and relationship with others). The sample was composed of 165 university students enrolled in courses related to physical activity and sport sciences, with a mean age of 20.33 years (SD = 3.44), studying in a Spanish public university located in Almeria, in the southeast of Spain (Andalusia). The main findings showed the existence of a continuous and effective relationship between self-efficacy and basic psychological needs. While there was no positive and direct relationship between cognitive anxiety and somatic anxiety and autonomy, there was a direct and positive relationship between cognitive anxiety and somatic anxiety and competence and the relationship with others. Therefore, the results obtained showed that a more positive form of motivation would be autonomous motivation, as it helps to interpret the perception of self-efficacy, favoring performance in competition, whereas controlled motivation has the opposite effect. The importance of this research resides in the fact that it shows that within the sports field, an athlete’s self-perception has an indirect negative effect on pre-competitive somatic anxiety, and the link in this relationship is task orientation and the absence of demotivation towards sport. Despite this, the same effect on cognitive anxiety was not produced.

## 1. Introduction

The main theoretical underpinning contribution to basic psychological needs is the self-determination theory (SDT) [1,2,3]. This theory [4] refers to the motives that lead a person to initiate and continue an action [5] or, in other words, to motivation and its types [6]. This theory is understood as a constant in which different levels of self-determination are established. Depending on the higher or lower degree of self-determination, an athlete’s behaviors can be intrinsically or extrinsically motivated [7]. If the athlete has a high degree of self-determination, it is because of his or her intrinsic motivation, and this leads to a greater commitment to and enjoyment of sport [8].

To increase people commitment, it is necessary that their basic psychological needs are satisfied; therefore, the theory of self-determination considers that autonomy, competence and the relationship with others motivate and drive human behaviors [9,10,11]. Consequently, performance is at its peak when people satisfy these needs, thus increasing their engagement, performance, and learning [8]. They are defined as “innate psychological nourishments that are essential for prolonged psychological growth, wholeness and well-being” [12]. In turn, these needs require the necessary conditions for psychological health, and in addition, their satisfaction is associated with more effective functioning [13]. Prior research indicates that each of these needs plays an important role in people development and experience, as well as in their well-being in everyday life [12], so that none of them can be neglected without negative consequences.

It is within these theoretical contributions that three needs are situated, namely, competence, autonomy, and relatedness to others [8]. In detailing each of them, we can say that:
(i)Competence refers to the way in which a subject establishes relationships and deals with their daily actions efficiently and with confidence [14], through which students feel capable of being able to adequately carry out everything they set out to do, that is, with the feeling of being able to solve the tasks they are given with a high probability of success and with great efficacy [15]. Students perceive their level of competence when they compare themselves with the rest of their peers and, depending on the results they obtain, rate the level they reached, according to ego orientation, as better or worse, instead of measuring their competence in terms of their personal success, comparing their results with those they themselves obtained previously, thus establishing their personal improvement [16]. Thus, when students focus exclusively on outperforming others by continually comparing themselves to others, the perception of their competence begins to be jeopardized and may become unreliable [17]. This can have negative consequences for students, as they can become demotivated and start to feel incompetent and to undervalue their own activities [12]. All of these issues make necessary the assessment of the motivational climate influenced by the teachers, since if the signals they project are intense, it may be that students’ competence orientation to carry out their tasks may be reduced or modified [18], damaging their perception of competence in their classes.(ii)Autonomy refers to the decisions that a subject makes in a self-sufficient manner, which refers to the feeling students have of being the protagonists and participants in what they do, being able to assess their own performance and make decisions to solve tasks [19]. This autonomy improves when students feel that their opinions are valued and respected, their feelings are considered, and they are given the opportunity to make their own decisions [12]. Some research relates autonomy support to two aspects of sport practice, namely, motivation and lifestyle [20]. (iii)Numerous studies have shown the importance of autonomy support for students, due to its positive influence on intrinsic motivation [21,22,23,24], leading to greater students’ engagement in learning and increased adherence to sport [25]. On the other hand, autonomy is also positively related to an active lifestyle, in which students perform physical exercise on a regular basis [26,27,28], thus allowing them to achieve benefits for their physical, mental, and social health [29,30]. Other authors showed that the autonomy support encouraged by teachers helps students to relate to others and favors a good performance in the required tasks [31].(iv)Relatedness to others refers to the type of interpersonal relationships that a subject establishes [32], since students need to feel connected with their teachers, classmates, and school [8,12]. (v)It should be mentioned that the social context is very important for students, as the attitude of both teachers and peers will influence the fulfilment of the other two basic psychological needs [33]. The most influential factor is a student’s relationship with the teachers, showing that a teacher promotes the opportunity for social connections between students during teaching [34,35]. It is also very important for students’ personal development in adolescence, as studies show that feeling socially connected is a much stronger predictor of self-determined motivation than the other two needs [36]. The literature shows that when students satisfy the three basic psychological needs described above, their engagement, achievement and learning increase, with the school being the primary place to achieve this [37].

Regarding anxiety, we start from the theoretical conception that establishes that different types of motivation have different affective, cognitive and behavioral inferences on students depending on their involvement in an activity, and consequences, such as negative emotions [12,38,39]. These have been functionally defined as anxiety in most cases [40]. Thus, we find competitive anxiety, defined as an impending emotional state characterized by feelings of apprehension, tension and nervousness, linked to elevated nervous system activity, generated by sporting competition [41], and when this anxiety occurs prior to competition, it is referred to as pre-competition anxiety [42].

Different theories interpret the symptoms of pre-competition anxiety as positive and not always negative [43]; thus, pre-competition anxiety can also be considered multidimensional, differentiating between intensity and direction [40]. If we mention intensity, it refers to the experienced level of precompetitive anxiety symptoms. On the other hand, if we mention direction, this refers to the extent to which students interpret the potency of the symptoms associated with pre-competitive anxiety to either favor or diminish their performance [44].

Some researchers included in their studies analyses to look for gender differences in pre-competition anxiety, indicating differences in the levels of cognitive anxiety and/or self-efficacy [45,46,47]. In contrast, other researchers, in different studies, did not show such differences [48]. All this may suggest that there may be external factors involved in the assessment of pre-competitive anxiety. Therefore, it should be approached as a successive phenomenon because of a series of antecedents [49]. In this sense, three components have been identified, i.e., somatic anxiety, cognitive anxiety and self-efficacy, from the theory of multidimensionality [49].

In short, when we talk about anxiety, we are referring to a negative emotional state consisting of feelings associated with nervousness, worry and apprehension, as well as with various aspects related to activation or arousal. They include a physical component (which is called somatic anxiety) and a mental component (which is called cognitive anxiety) [50]. However, through the creation of the anxiety measurement instrument that we used in this research [49], we divided cognitive anxiety into two subcomponents: one consisting of the items considered positive, which was called the state of self-confidence, and the other consisting of items with a more negative interpretation, called cognitive anxiety. Therefore, somatic anxiety is a state in which the subject’s thoughts and sensations are manifested at the body level, for example, in difficult breathing, palpitations, sweating, etc. Cognitive anxiety is a state in which the subject has distressing and negative thoughts that significantly affect performance and attention. Self-confidence is postulated as the opposite of cognitive anxiety, that is, it is that state of anxiety that is not negative, but rather somehow drives the subject to face a challenge.

Within sport psychology, there are numerous investigations that aimed to study the different types of anxiety that athletes can suffer [50,51]. Anxiety is understood as an emotional state characterized by a feeling of apprehension and tension [52,53], while other authors define it [54,55,56] as a subjective feeling of a perceived threat, sometimes accompanied by an increase in physiological activation and understood as an emotional state of negative character in which the athlete feels nervousness, tension and apprehension associated with the activation of the body’s arousal system [50].

Research has shown that these emotions appear in pre-competitive situations [57], where the concept of pre-competitive anxiety (PCA) arises, which can have both a positive and a negative effect on the athlete and has been consolidated within sports psychology as a topic of great interest [58,59]. Its manifestations are usually on a somatic level through excessive sweating, trembling of the limbs or increased heart rate, but also on a psychological level, through paralyzing fear, mental dispersion, reduced self-esteem and the possible increase in frustration or guilt [60]. The increase in research on this topic has allowed the implementation of different techniques by sports psychologists, focusing on teaching fear coping, visualization, goal setting, relaxation or the improvement of self-efficacy [61].

Taking into consideration the above theoretical contributions, the impetus for the choice of the sample for this research was the desire to investigate the situation of students and future professionals of physical activity and sport sciences. To this end, we analyzed their self-perceptions of both anxiety and basic psychological needs. Therefore, the aim of this work was to investigate the relationship between anxiety (somatic anxiety, cognitive anxiety and self-efficacy) and basic psychological needs (competence, autonomy and relationship with others).

Our aim was to answer the following research questions that guided this study: what kind of relationship exists between anxiety and basic psychological needs? Is there the same relationship between the different types of anxiety and the different basic psychological needs? Is it possible to establish the existence of a direct and positive relationship between both psychological constructs or, on the contrary, is there a negative relationship between them? For this, the following were considered as variables for anxiety: levels of somatic anxiety, cognitive anxiety and self-efficacy; for the basic psychological needs, we considered competence, autonomy and relationships with others. As the study population, we chose a total of 165 undergraduate and master’s degree students of sciences of physical activity and sport who were studying at a Spanish public university located in the southeast of Spain, Andalusia, specifically, in Almeria.

The hypotheses of this work were as follows:
**Hypothesis** **1.***There is a continuous and effective relationship between self**-efficacy and basic psychological needs.*
**Hypothesis** **2.***There is a negative and objective relationship between cognitive anxiety and basic psychological needs.*
**Hypothesis** **3.***There is a negative and objective relationship between somatic anxiety and basic psychological needs.*

## 2. Materials and Methods

### 2.1. Participants

The sample consisted of 165 university students [70.9% (*n* = 117) men and 27.9% (*n* = 46) women], with a mean age of 20.33 years (SD = 3.44), studying physical activity and sports sciences (both for a degree in physical activity and sports sciences and for a master’s degree in research in physical activity and sports sciences), who were studying at a Spanish public university located in the southeast of the Spanish country, specifically, in Andalusia, in the city of Almería.

It should be noted in this section that in the Spanish university system, the studies leading to the teaching of physical education are divided into three large areas: the first of them refers to the official degree in physical education and sports sciences, the second refers to a specific mention in physical education within the studies of the official degree in primary education, and the third refers to postgraduate or master’s studies, which lead to both an official master’s degree in physical activity and sports sciences as well as a master’s degree in teacher training with a mention in physical education.

In this case, this research considered students in the three large areas of university training for teaching physical education, with a distribution of the sample as follows: 76.7% (*n* = 147) of the students were studying either for the official degree in physical activity and sports sciences or for the official degree in primary education with a mention in physical education, while 23.3% (*n* = 18) of the students were enrolled in postgraduate studies related, in this case, to the official master’s degree in physical activity and sport sciences and the master’s degree in teacher training with a mention in physical education.

Regarding the marital status, 1.8% (*n* = 3) of the sample were married or living with a partner, while 98.1% (*n* = 162) of it stated that they were single. On the other hand, the sample was also asked about the weekly hours dedicated to practicing sports in their free time. Thus, 64.8% (*n* = 107) of the participants practiced sports for between 3 and 6 hours a week, 20% (*n* = 48) practiced sports for more than 6 hours a week and 15.38% (*n* = 10) practiced sports for a maximum of 3 hours a week (See Table 1).

The type of sampling used was simple random. As inclusion criteria, to participate in the study, the students had to be enrolled in some of the undergraduate or master’s courses related to the sciences of physical activity and sport, had to be of legal age (18 years or older) and had to participate in the study voluntarily. As exclusion criteria, we discarded those questionnaires with missing answers. The sample size was calculated using the Soper’s a priori sample size calculator for structural equation models [62]. We obtained an expected effect size of 0.30, a probability level of 0.05 and a desired power level of 0.95, and the minimum recommended effect size was 200 cases. This result suggests that the ideal sample size corresponded to a number approximating the total number of participants that eventually were in our research sample (Figure 1).

The N of the universe of the study understood as the set from which the information was extracted was 277 participants, taking as a guide to calculate it, the total number of students enrolled in each of the four courses of the official degree in sciences of physical activity and sport, the total number of students enrolled in the mention in physical education of the official degree in primary education. the total number of students enrolled in the master’s degree in physical activity and sport sciences and the total number of students enrolled in the specialization in physical education within the official master’s degree in teacher training.

The data collection procedure was as follows:

In the first place, we designed the temporary and theoretical planning of the investigation. In this case, it was an empirical study based on the application of a quantitative, descriptive and cross-sectional methodology.

After its design, the project was submitted for evaluation to the Bioethics Committee of the University of Almería, being approved by the Institutional Review Board of the University of Almería (UALBIO2022/035) given that the Helsinki guidelines were met (declaration on ethics of the investigation).

After obtaining approval, we contacted each of the teachers responsible for the subjects to which the questionnaires were intended to be administered. In this case, we selected a teacher for each course and we contacted them by email, providing information about the purpose of the study and requesting their permission to come to the classroom on the agreed day and time.

Once the teachers agreed and gave us 20 minutes at the beginning or end of their class, we went to the classroom to administer the questionnaire. This was carried out during the first four-month period of the 2021/2022 academic year in each of the four courses of the official degree in physical activity and sports sciences, in the mention in physical education of the official degree in primary education, in the official master’s degree in physical activity and sports sciences and in the specialization in physical education of the official master’s degree in teacher training.

During this process, the principal investigator was present to ensure the willingness of the students to participate in the study and obtain their informed consent, as well as to offer brief information on the purpose and instructions for completing the questionnaire, taking basic psychological needs and anxiety as variables. Before beginning, the anonymity of the responses and the confidentiality of the data were guaranteed, spending around 20 min on their responses, without any student reporting problems in completing it.

### 2.2. Ethical Procedures

This study was conducted in accordance with the Declaration of Helsinki and was approved by the Institutional Review Board of University of Almería (UALBIO2022/035). As for the procedure, a date and time were previously agreed with the lecturer responsible for the subject, to be able to attend his or her class. Once there, we included only those students enrolled in the corresponding course and subject who were informed of the data protection protocol, signed the form to give their consent to participate in this research and completed the questionnaires.

### 2.3. Instruments

To assess the anxiety levels, we used the CSAI-2 [49] Spanish version [63], later revised and used by other researchers [64,65], which measures the levels of somatic anxiety, cognitive anxiety and self-efficacy. It is composed of 27 items evaluated on a Likert-type scale (1–4). The psychometric properties of this instrument are appropriate in terms of reliability (Cronbach’s alpha) (cognitive anxiety: α = 0.87; self-efficacy: α = 0.93; somatic anxiety: α = 0.90) [66]. We calculated the internal consistency score and obtained an α = 0.84. This instrument corresponds to a standardized scale validated by other authors [63,64,65].

The second instrument used was the Basic Psychological Needs in Exercise Scale [67], Spanish version, adapted to physical education [68]. It is composed of 12 items evaluated on a Likert-type scale (1–5), measuring competence (4 items), autonomy (4 items) and relationship with others (4 items). The psychometric properties of this instrument are appropriate in terms of reliability (Cronbach’s alpha) (autonomy: α = 0.77; competence: α = 0.80; relatedness to others: α = 0.89) [68]. We calculated the internal consistency score and obtained α = 0.90 for autonomy, α = 0.73 for competence, and α = 0.76 for relatedness. This instrument corresponds to a standardized scale validated by other authors [68].

The answers to these two questionnaires were combined with certain socio-demographic data, such as age, sex, marital status, hours spent playing sport, current course, or qualification; we prepared a booklet in which we presented the instruments to the participants.

### 2.4. Data Analysis

The first statistical analyses were descriptive, calculating the mean, standard deviation and bivariate correlations. This was followed by reliability scores. Regarding variance, we calculated the hierarchical omega [69] and the common explained variance (ECV) [70]. Hierarchical omega scores ≥0.70 indicate the presence of a unidimensional structure [70], scores below 0.70 indicate multidimensionality, and values above 0.85 indicate one dimensionality [69]. Also at the item level, the ECV-I [69] was calculated to identify the variance scores of each item explained by the FG; scores ≥0.80 indicate a significant influence of GFR [70]. Ultimately, values >0.70 indicate the fact that a latent variable is adequately defined by its indicators [70].

The following statistical analyses were carried out using a second-order structural equation model (SEM). The reason for the choice of this test was that in addition to interpreting the structure at a higher level and its effects on certain dependent variables, it provided us with a congruence to deal with certain forms of multicollinearity [71,72,73].

To verify normality, we used the Kolmogorov–Smirnov test and obtained a score of 0.36. Based on this result, we rejected the null hypothesis that there was no difference between the means and we concluded that there was a significant difference.

Finally, to consider the model valid, we took into account the scores obtained for different indices, i.e., the TLI (Tucker–Lewis index), SRMR (standardized root-mean-square residual) and RMSEA (root-mean-square error of approximation). TLI values above 0.95, SRMR values below 0.06 and RMSEA values below 0.08 were considered adequate [74].

All these tests were carried out using the SPSS program (version 26), the statistical analysis program R (version 2015) and the analysis modules belonging to the “Lavaan” package.

## 3. Results

Table 2 presents the scores obtained in relation to somatic anxiety, cognitive anxiety and self-efficacy, as well as autonomy, competence and relationships with others.

The correlations that are presented in the following table measure the relationships t established between the different variables. They had values between −1 and +1. However, the further the score is from zero, the stronger the relationship between the two variables. In this case, all correlations in the table are positive.

Below are a series of acronyms that correspond to the following variables: PE1, autonomy; PE2, competition; PE3, relationships with others; ACG, cognitive anxiety; ATC, self-efficacy; ASS, somatic anxiety.

In addition, the hypothetical model of predictive relationships (Figure 2) provided the following values for its various indices.

The overall fit indices (they evaluate the model in general) were adequate: *p* < 0.000, GFI = 0.946. 

Incremental or comparative fit indices (compare the proposed model with the model of independence or absence of relationship between the variables), CFI = 0.866; IFI = 0.875.

The relationships established in the structural equation model showed that: There was no positive and direct relationship between cognitive anxiety and autonomy;There was no positive and direct relationship between somatic anxiety and autonomy;Autonomy and self-efficacy were positively correlated (=0.06, *p* < 0.001);Cognitive anxiety and competence were positively correlated (=0.06, *p* < 0.001);Cognitive anxiety and relatedness were positively correlated (=0.07, *p* < 0.001);Somatic anxiety and competence were positively correlated (=0.04, *p* < 0.001);Competence and self-efficacy were positively correlated (=0.06, *p* < 0.001);Somatic anxiety and relatedness were positively correlated (=0.06, *p* < 0.001);Self-efficacy and relationships with others were positively correlated (=0.06, *p* < 0.001).

Hypothesis 1 of this work is fulfilled: there was a continuous and effective relationship between self-efficacy and basic psychological needs. Hypotheses 2 and 3 were partially met, since there was no positive and direct relationship between cognitive anxiety and somatic anxiety and autonomy, but there was a direct and positive relationship between cognitive anxiety and somatic anxiety and competence and relationship with others. Therefore, this study provides evidence as regards college sports that athlete self-perception has an indirect negative effect on precompetitive somatic anxiety, and the link in this relationship is task orientation and lack of motivation towards sport. However, the same effect was not found for cognitive anxiety. This means that if these athletes perceive teamwork, value learning and define themselves as competent according to self-referential criteria and do not lack the intention to practice, they present fewer symptoms of increased nervous system activation (e.g., sweaty hands, muscle tension, increased heart rate) before a competition. However, this motivational climate does not confirm that negative thoughts and images about pre-competition performance decrease when an athlete is task-oriented and lacks motivation.

## 4. Discussion

The objective of this study was to analyze the relationship between anxiety and basic psychological needs in students of physical activity and sports sciences. This objective was met. The main findings of this work allow us to affirm the existence of a continuous and effective relationship between self-efficacy and basic psychological needs [5,6,7]. However, the same type of relationship does not hold if we focus on cognitive anxiety and somatic anxiety and autonomy [14], despite the existence of a direct and positive relationship between cognitive anxiety and somatic anxiety and competence and relationships with others, as previously demonstrated by research [4,13,21].

The main theoretical implication of this research lies in the importance that the athlete’s self-perception acquires and the indirect negative effect it has on precompetitive somatic anxiety [42,43,44,45,46,47,48,49,50,51,52]. However, this contribution also has implications at a practical level, since our findings regarding both the orientation to the task and the lack of motivation towards sport are among the great practical contributions of this work. That is, the importance that motivation [9,16] and the motivational climate have in practice, so that if an athlete is task-oriented and motivated, he defines himself as competent according to self-referential criteria, with fewer symptoms of great nervous system activation (e.g., sweaty hands, muscle tension, increased heart rate) before a competition [53,54,55,56,57,58].

According to the contributions of various authors [18,21,36], we are faced with a situation characterized by the climate of task involvement having an indirect negative effect on somatic anxiety and no indirect effect on cognitive anxiety. It may be that, as the literature suggests in the context of sport, somatic anxiety is a conditioned response to environmental stimuli (e.g., motivational climate), while cognitive anxiety would be related to personal factors such as perceived ability [49]. Therefore, it is necessary to promote a sports practice focused much more on the process than on the result, which would allow the athlete to face the challenge of the competition as something positive or conducive to his good performance in the competition [42,44,53]. All this translates into an additional positive effect, with the athlete intensifying his efforts to achieve his goal, strengthening his perception of competence and controlling negative emotions effectively [75].

Therefore, the results obtained in our research show that a more positive form of motivation is autonomous motivation, since it helps to interpret the perception of self-efficacy as favorable for performance in competition, while controlled motivation has the opposite effect. These results also lead us to consider subjects related to sports psychology for the training that we offer our students, being sports psychology committed to helping athletes becoming involved in practical sport in different ways, by their own will, because they value it or because they directly perceive a climate of involvement in the task, but also indirectly. In this case, the link is provided by the definition of self-referred competence (for example, through learning and focusing on improving day by day through effort). In short, although there are numerous studies that evaluated the mediating effect of goal orientations on the relationship between motivational climate and motivational regulations [64], they considered separate motivational regulations, while the present work tested it with the three types of motivation proposed by TAD, supporting previous results [76] according to which autonomous motivation is more dependent on personal reasons and is facilitated by contextual events. In general, the results support previous conclusions [77,78,79], i.e., the fact that the psychological climate or environment is a critical factor to predict the cognitive components of motivation.

## 5. Limitations and Future Directions

It should be noted that this study has limitations, such as the use of a cross-sectional research design, which did not allow causal relationships between variables to be established. It would be interesting to conduct future longitudinal studies to understand in greater depth the possible applied implications that this model may have in the sporting context, given that different types of competitions make anxiety relatively unstable. 

Another limitation is the small sample size, which suggests the need for future research to replicate this study with a larger sample and considering different levels of competition, as well as to evaluate the effect of the perception of competence in these associations, in addition to testing the models based on gender, due to the differences that self-efficacy tends to show in different genders.

## 6. Conclusions

In conclusion, this study partially confirmed the relationships hypothesized in the proposed models. Thus, for success in contexts such as university sport, the climate of involvement in the task tends to produce improvements in the interpretation of positive emotions prior to a competition, if the success criterion is self-referred and the behaviors towards sport practice are autonomous; it also reduces the symptoms of somatic anxiety when the success criterion is self-referred and a lack of intention to practice is absent.

## Figures and Tables

**Figure 1 ijerph-20-02415-f001:**
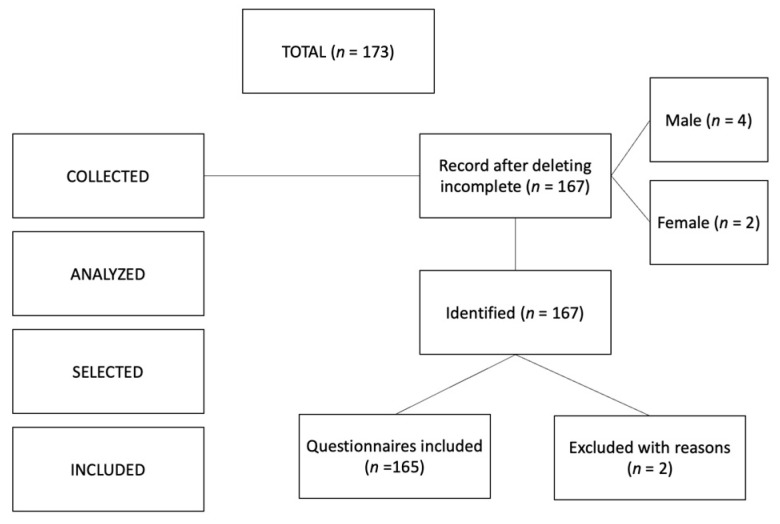
Flowchart of the sample selection procedure.

**Figure 2 ijerph-20-02415-f002:**
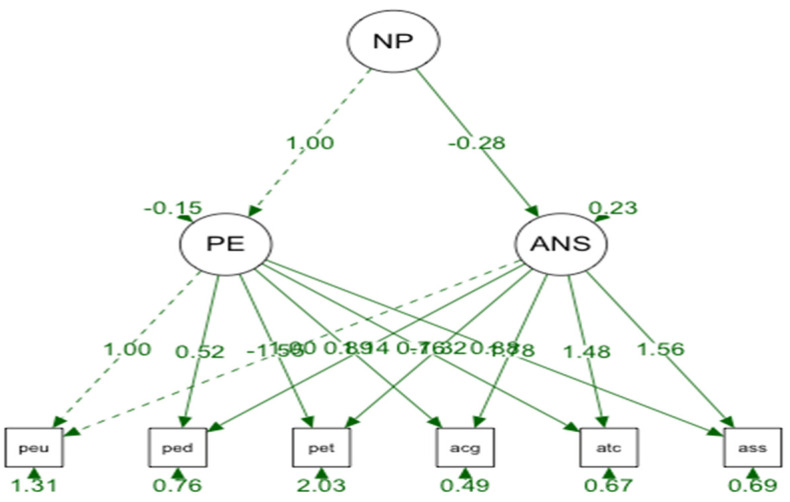
Structural equation model.

**Table 1 ijerph-20-02415-t001:** Description of the sample according to course and sex.

	Females	Males	Total
First course	18 (39.1)	68 (58.1)	86
Second course	16 (34.8)	23 (19.7)	39
Third year	6 (13%)	14 (12%)	20
Total	40	105	145
Master’s degree	6 (13%)	12 (10.3)	18
Total	46	117	163

**Table 2 ijerph-20-02415-t002:** Preliminary analysis.

	PE1	PE2	PE3	ACG	ATC	ASS
PE1		0.237 **	0.361 **	0.142	0.493 **	0.053
PE2			0.502 **	0.007	0.191 *	0.008
PE3				0.000	0.283 **	0.049
ACG					0.031	0.601 **
ATC						0.102
ASS						

Note. * *p* < 0.05; ** *p* < 0.01.

## Data Availability

The data is not publicly available due to ethical or privacy restrictions, however, it may be available if requested from the corresponding author.

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
