# Peer review of "Levels of Somatic Anxiety, Cognitive Anxiety, and Self-Efficacy in University Athletes from a Spanish Public University and Their Relationship with Basic Psychological Needs"

_ijerph, 2023, doi:10.3390/ijerph20032415_

Round 1

Reviewer 1 Report

I would like to thank you for the opportunity since I feel very fortunate to be able to review this article and I would like to congratulate the authors for this work. For me, as an educator, this topic is very important and has a lot of value. I detail my suggestions below and in concluding my consideration.

This manuscript investigated the association between somatic anxiety, cognitive anxiety, and self-efficacy) and basic psychological needs (competence, autonomy, and relatedness).

Title: The title is concrete, representative and indicative of the problem investigated in the manuscript. As a suggestion, the title should provide information about the place where the research was conducted.

Abstract: The abstract is clear and complies with the general rules for writing a good abstract. However, I would like to see a better description of the sample, indicating the context.  This is the most important section of the paper since it will be read many more times than even the manuscript itself, so it needs the most attention. A brief note on the importance of the research is an excellent ending to a high-level abstract.

Introduction

As I mentioned, I find this research extremely important in contributing to the field of psychology and education. I do not disagree with the authors' justifications and read many very good and current arguments.  

It is suggested to the authors that based on the stated objective they highlight the research questions that help to conduct the research and discussion based on the findings found in which the study variables, the study population, and the expected outcome appear.

Material and method.

Participants. This section should be better defined. It is suggested that the inclusion criteria be better specified. It is also suggested to report which studies were selected that were related to physical activity and sport sciences (Physical Education Teaching? Sport Sciences?). Specify better which degrees are related within the current educational system in the country. It is suggested to better specify the context, country, region, etc. It is also suggested, since a sample calculation is reported, to report the N of the universe to be able to corroborate that 200 cases would be sufficient. 

The instrument section indicates that data were collected related to sociodemographic variables that are not specified in this section, is there a reason?

Instruments: It is suggested to include information about whether the instruments were already validated in the target population previously. 

Statistical analysis. It is suggested to include information about whether the distribution of the data complied with the assumption of normality. It is also suggested to include references about the interpretation of the omega values (line 201).

Results: 

The results are correctly displayed. It is suggested to include information prior to the results in Table 2 to help readers unfamiliar with this type of analysis or methodology to understand the results.  

Discussion: It is suggested that this section be rewritten. It seems more like a presentation of results than a discussion of the findings with other studies. In fact, much of the information in this section is what is missing in the results section. It is suggested that a section on practical and theoretical implications be included to evaluate the scope of the research.

Conclusions: They are clear and provide an answer to the stated objectives. 

I suggest that the authors be allowed to resubmit this manuscript after a second round of review. I will be happy to read it again.

Author Response

REVIEWER 1

  • I would like to thank you for the opportunity as I feel very fortunate to be able to review this article and I would like to congratulate the authors for this work. For me, as an educator, this topic is very important and has a lot of value. I detail my suggestions below and at the conclusion of my consideration. This manuscript investigated the association between somatic anxiety, cognitive anxiety, and self-efficacy) and basic psychological needs (competence, autonomy, and relationship).

Dear reviewer, thank you very much for your contributions which substantially help to improve this work. We appreciate the advice and time spent on this work. All changes made are indicated in green.

  • Title: The title is concrete, representative and indicative of the problem investigated in the manuscript. As a suggestion, the title should provide information about the place where the research was conducted.

Thank you very much for your contribution. We have included in the title that it is a study carried out in a Spanish public university.

  • Summary: The summary is clear and meets the general rules for writing a good summary. However, I would like to see a better description of the sample, indicating the context. This is the most important section of the document, as it will be read many more times than even the manuscript itself, so it needs the most attention. A brief note on the importance of research is an excellent end to a high-level summary.

A brief description of the context in which the research was carried out has been added to the summary, specifically, in a Spanish public university, located in the southeast of the Spanish country, located in Andalusia, and in the city of Almería specifically (lines 19-20).

Likewise, the summary ends by indicating the importance of this research and its implications in the sports field (lines 27-30).

  • Introduction As I mentioned, I find this research extremely important to contribute to the field of psychology and education.

Thank you very much for your comment.

  • I do not disagree with the authors' justifications and read many very good and current arguments.

Thank you very much for your comment.

  • It is suggested that the authors based on the objective set highlight the research questions that help to carry out the research and discussion based on the findings found where the study variables, the study population and the expected outcome appear.

We have added several research questions that drive this manuscript. As well as the variables that have been taken into account in the study of psychological constructs such as anxiety and basic psychological needs. In addition, the study population is specified. Regarding the expected outcome, a total of three hypotheses of this research are formulated in this regard (lines 198-216).

Material and method.

  • This section should be better defined.

We have improved the description of the sample, this section has been rewritten (lines 220-140).

  • It is suggested that the inclusion criteria be better specified.

This information about the sample inclusion criteria has been included (lines 256-260)

  • It is also suggested to report which studies were selected related to physical activity and sport sciences (Teaching Physical Education? Sports Science?). Better specify which careers are related within the current education system of the country.

A brief explanation has been added about the organization of the studies leading to the teaching of physical education within the Spanish university system, as well as the distribution of the sample in each of them in this research. As well as each of the degrees that refer to it have been identified (lines 241-252).

  • It is suggested to better specify the context, country, region, etc.

We have added this information in the sample description (lines 224-225)

  • It is also suggested, given that a sample calculation is reported, to report the N of the universe in order to corroborate that 200 cases would be enough.

The N of the universe calculated according to the students enrolled in each of the four courses of the official degree in physical activity and sport sciences has been added. The total number of students enrolled in the mention in physical education of the official degree in primary education. The total number of students enrolled in the master's degree in physical activity and sport sciences And, the total number of students enrolled in the specialization in physical education within the official master's degree in teacher training (lines 267-274).

  • The instrument section indicates that data related to sociodemographic variables that are not specified in this section were collected, is there a reason?

We have added the information referring to marital status and weekly hours of sports practice in leisure time per week provided by the sample (lines 248-252).

  • Instruments: It is suggested to include information on whether the instruments were already validated in the target population previously.

This information has been added. Both instruments correspond to previously validated and standardized scales that have been carried out within different investigations developed in the field of sports psychology (lines 326-327 / 334-335).

  • Statistical analysis. It is suggested to include information on whether the distribution of data met the assumption of normality.

To contrast normality, we used the Shapiro-Wilk test, where we obtained a score of .027. From this result we reject the null hypothesis that there is no difference between the means and conclude that there is a significant difference (lines 354-356).

  • It is also suggested to include references on the interpretation of omega values (line 201).

A total of three references from the years 2020 and 2022 have been included that support the interpretation of omega values (line 345).

  • Results: The results are displayed correctly.

Thank you very much for your comment.

  • It is suggested to include information prior to the results in Table 2 to help readers unfamiliar with this type of analysis or methodology understand the results.

An explanation of the scores represented in Table 2 has been added.

They measure the relationships that are established between the different variables. They are between -1 and +1. However, the farther the score is from zero, the stronger the relationship between the two variables. In this case, all correlations shown in the table are positive (lines 367-371).

A legend indicating the meaning of * has also been added in the table scores (line 415).

  • Discussion: It is suggested to rewrite this section. It seems more like a presentation of results than a discussion of the findings with other studies. In fact, much of the information in this section is missing from the results section.

Both the results section (lines 505-519) and the discussion section (lines 521-569).

  • It is suggested to include a section of practical and theoretical implications to evaluate the scope of the research.

The discussion has been rewritten obeying the following sections: objectives achieved, main findings, theoretical and practical implications and educational implications of the findings obtained (lines 521-569).

Conclusions: They are clear and respond to the objectives set. I suggest that authors be allowed to resubmit this manuscript after a second round of review.

I'll be happy to read it again.

Thank you very much again for your time and suggestions. We hope that everything is correctly addressed and suitable for acceptance

Reviewer 2 Report

Dear authors.

I would like to make some formal and substantive comments which, in the opinion of this reviewer, would improve your interesting contribution.

The space in the term "self- efficacy" between "-" and "efficacy" needs to be removed throughout the document. There also seems to be a formal oversight in line 194 (use of "0." in some cases and not in others).

In the introductory chapter, given the later relevance of these concepts, it would be highly recommendable to define in an explicit way (or to indicate concretely with what content provided corresponds to their meaning) "somatic anxiety", "cognitive anxiety" and "self-efficacy" from the theory of multidimensionality. I understand that this conciseness would favour the dissemination of the content among non-specialist readers.

In the section on materials and methods, it is necessary to indicate in greater detail the procedure for obtaining the data, beyond the mention of this in rows 164 and 165.

In the results section, in order to facilitate the consultation of the data, it would be highly advisable to facilitate the identification of PE1, PE2, PE3, ACG, ATC and ASS with the variables somatic anxiety, cognitive anxiety and self-efficacy, and autonomy, competence, and relationships with others. This observation does not only refer to the reading of the data in table 2 but can also be extended to the consultation of the data in figure 2.

It would be necessary to clarify the meaning of the "**" mark that appears in some of the data in table 2.

It would be highly advisable to explain the scores given in table 2 in order to make them easier to understand for readers unfamiliar with the instruments used.

Reference number 56 needs to be included in the reference section.

Kind regards.

Author Response

REVIEWER 2

  • Dear authors. I would like to make some formal and substantive comments that, in the opinion of this reviewer, would enhance his interesting contribution.

Dear reviewer, thank you very much for your contributions which substantially help to improve this work. We appreciate the advice and time spent on this work. All changes made are indicated in green.

  • The space in the term "self-efficacy" between "-" and "effectiveness" should be removed throughout the document.

Removed the use of "-" in these words

  • There also seems to be a formal oversight on line 194 (use of "0." in some cases and not in others).

This error has been corrected throughout the manuscript.

  • In the introductory chapter, given the subsequent relevance of these concepts, it would be highly recommended to explicitly define (or specifically indicate with what content contributed corresponds to their meaning) "somatic anxiety", "cognitive anxiety" and "self-anxiety". -effectiveness" of the theory of multidimensionality. I understand that this conciseness would favor the dissemination of the content among non-specialist readers.

A brief explanation of what we mean by somatic anxiety has been added: that the thoughts and sensations of the subject are manifested at the body level, for example difficulty breathing, palpitations, sweating ... Cognitive anxiety: one in which the subject has distressing and negative thoughts that significantly affect performance and attention. And self-confidence: it is postulated as the opposite of cognitive anxiety, that is, it is that state of non-negative anxiety, but somehow drives the subject to face a challenge (lines 155-168).

  • In the section on materials and methods, it is necessary to indicate in greater detail the procedure for obtaining the data, beyond the mention of this in rows 164 and 165.

The procedure for obtaining the data in detail has been detailed, from the design of the research itself, to the completion of the questionnaires by the sample (lines 290-311).

  • In the results section, to facilitate the consultation of the data, it would be highly recommended to facilitate the identification of PE1, PE2, PE3, ACG, ATC and ASS with the variables somatic anxiety, cognitive anxiety and self-efficacy, and autonomy, competence and relationships with others. This observation not only refers to the reading of the data in Table 2 but can also be extended to the query of the data in Figure 2.

This information has been added before Table 2 in order to clarify the acronyms that appear below in the manuscript, both in Table 2 itself, and in Figure 2: (lines 410-412)

PE1: autonomy

PE2: competition

PE3: relations with others

GCA: cognitive anxiety

ATC: self-efficacy

SSA: somatic anxiety

  • The meaning of the mark "**" in some of the data in Table 2 should be clarified.

Below table 2 a note clarifying this meaning has been added (line 415).

  • It would be highly recommended to explain the scores given in Table 2 so that they are more understandable to readers who are not familiar with the instruments used.

An explanation of the scores represented in Table 2 has been added.

They measure the relationships that are established between the different variables. They are between -1 and +1. However, the farther the score is from zero, the stronger the relationship between the two variables. In this case, all correlations shown in the table are positive (lines 367-371).

  • Reference number 56 should be included in the reference section.

Reference 56 (lines 803-804) has been included

Round 2

Reviewer 1 Report

I am deeply grateful to the authors for taking my suggestions into consideration. 

Although I consider that the article has improved considerably, is more solid, clearer and could be accepted in IJERPH if the editors so consider, I would like you to clarify why Shapiro Wilk was used to explore whether the assumption of normality in the distribution of the data was fulfilled and not Kolmogorov Smirnov, which I consider more correct since the sample was greater than 50.

Thank you very much.

Author Response

Dear reviewer,

Taking their contributions into account, we have realized that it is due to an error on our part, since from the theoretical contributions it is recommended to carry out the Kolmogorov-Smirno test when the study sample is greater than 50 subjects. Therefore, we have carried out said test, obtaining a score of .036. You can find this information on lines 354-356.